# The Combination of Citrus Rootstock and Scion Cultivar Influences *Trioza erytreae* (Hemiptera: Triozidae) Survival, Preference Choice and Oviposition

**DOI:** 10.3390/insects15050363

**Published:** 2024-05-16

**Authors:** María Quintana-González de Chaves, Nancy Montero-Gomez, Carlos Álvarez-Acosta, Estrella Hernández-Suárez, Aurea Hervalejo, Juan M. Arjona-López, Francisco J. Arenas-Arenas

**Affiliations:** 1Unidad de Protección Vegetal, Instituto Canario de Investigaciones Agrarias (ICIA), Ctra. El Boquerón s/n, 38270 La Laguna, Spain; mquintana@icia.es (M.Q.-G.d.C.); nmonterogomez@gmail.com (N.M.-G.); 2Departamento de Producción Vegetal en Zonas Tropicales y Subtropicales, Instituto Canario de Investigaciones Agrarias (ICIA), Ctra. El Boquerón s/n, 38270 La Laguna, Spain; calvarez@icia.es; 3Department of Agri-Food Engineering and Technology, Andalusian Institute of Agricultural and Fisheries Research and Training (IFAPA), “Las Torres” Center, Ctra. Sevilla-Cazalla de la Sierra km. 12.2, 41200 Alcalá del Río, Spain; aurea.hervalejo@juntadeandalucia.es (A.H.); juanm.arjona@juntadeandalucia.es (J.M.A.-L.); fjose.arenas@juntadeandalucia.es (F.J.A.-A.)

**Keywords:** african citrus psyllid, citrus breeding programmes, Huanglongbing, tolerant plant material

## Abstract

**Simple Summary:**

Huanglongbing (HLB) is the most devastating citrus disease across Africa, America and Asia. It is associated with three species of bacteria belonging to the genus ‘*Candidatus* Liberibacter’, transmitted by the psyllid vector *Trioza erytreae*, among others. This insect has spread throughout part of the Iberian Peninsula, threatening citrus orchards with the possible arrival and transmission of HLB. Currently, there are no curative methods of control for the disease, so finding plant material less suitable for psyllids can reduce the risk of spreading. We have evaluated how the citrus combination, rootstocks (Flying dragon, ‘Cleopatra’ mandarin, Carrizo citrange, Forner-Alcaide no.5, Forner-Alcaide no.517 and *Citrus macrophylla*) and scion cultivars (‘Star Ruby’, ‘Clemenules’, ‘Navelina’, ‘Valencia Late’, ‘Fino 49’ and ‘Ortanique’) determine the biology of *T. erytreae.* Our results show that *T. erytreae* behaves differently depending on the combination, and is the least suitable for its survival with ‘Ortanique’ on Forner-Alcaide no. 517 and ‘Valencia Late’ on Carrizo citrange. The psyllid preferred ‘Fino 49’ grafted onto Carrizo citrange for egg laying rather than Flying Dragon.

**Abstract:**

*Trioza erytreae* (Del Guercio, 1918) (Hemiptera: Triozidae) is a citrus pest which produces gall symptoms on leaves and transmits bacteria associated with the citrus disease Huanglongbing, ‘*Candidatus* Liberibacter’ spp. In the present work, the biology and behaviour of *T. erytreae* were studied in different rootstock–cultivar combinations. Six rootstocks were used, Flying dragon (FD), ‘Cleopatra’ mandarin (CL), Carrizo citrange (CC), Forner-Alcaide no.5 (FA5), Forner-Alcaide no.517 (FA517) and *Citrus macrophylla* (CM), and six scion cultivars: ‘Star Ruby’, ‘Clemenules’, ‘Navelina’, ‘Valencia Late’, ‘Fino 49’ and ‘Ortanique’. Survival and oviposition were evaluated in a no-choice trial, and preference in a choice trial, all of them under greenhouse conditions. *Trioza erytreae* did not show a clear settle preference for any citrus combination. However, it was able to lay more eggs in ‘Fino 49’ grafted on CC than on FD. In terms of survival, ‘Ortanique’ grafted onto FA5 was more suitable than when grafted onto FA517, and in the case of ‘Valencia Late’, when it was grafted onto CM rather than CC. Our results showed that *T. erytreae* behave differently depending on the citrus combination.

## 1. Introduction

Spain, specifically the Mediterranean region, produces 5.6 million tonnes of citrus fruits, and is the fifth-largest citriculture in the world and the leading producer of the European Union [1]. The Spanish citrus sector has been threatened since 2014, when *Trioza erytreae* (del Guercio, 1918) (Hemiptera: Triozidae) was detected for first time in the Iberian Peninsula [2].

*Trioza erytreae* is a polyvoltine species with more than eight generations per year [3] whose females can lay up to 828 eggs [4,5]. Although the nymphs of this psyllid produce a typical gall symptom on leaves as a result of feeding [6,7], the major concern is that *T. erytreae* is one of the vectors of the bacteria species associated with the most destructive citrus disease, known as Huanglongbing (HLB) [8,9,10,11]. In recent decades, the citrus industry has been disrupted by the rapid and aggressive spread of HLB [8,12,13]. This disease is associated with the non-culturable Gram-negative bacteria species ‘*Candidatus* Liberibacter africanus’ (CaLaf), ‘*Ca*. L. americanus’ (CaLam) and ‘*Ca*. L. asiaticus’ (CaLas) [8,14,15,16,17]. These bacteria are restricted to the phloem vessels of plants [18], and to the gut, gut lining, haemolymph and cytosol of epidermal cells of the head, esophagus and salivary glands of vector psyllids [19,20,21,22]. This disease is distributed over a wide geographical range, across Africa, America and Asia, and so far has never been detected in Europe [23].

HLB disease is often represented as a tritrophic complex, where not only is the presence of the bacterium essential, but also the psyllid vector (capable of acquiring and inoculating it) and a suitable host plant for both the vector and the bacterium. Additionally, environmental conditions, especially temperature, play an important role in infection and symptom expression [8,24,25]. Most efforts in disease control associated with ‘*Ca.* Liberibacter’ bacteria have been directed at vectors: the evaluation of different trapping methods [26,27], insecticides [28,29] and biological control [30,31]. However, an integrated strategy should also focus on host plants.

Citrus breeding programs look for rootstocks adapted to certain abiotic stresses (salinity, limestone, flooding, drought, freeze and/or boron toxicity), or able to cope with some pests (*Phytophthora* spp., nematodes, Citrus Tristeza Virus (CTV), etc.) [32,33], which in combination with the scion cultivar obtain high productivity and good fruit quality. Citrus tolerance to HLB and its vectors is increasingly being considered by breeders [34,35,36,37,38,39], due to the absence of effective curative treatments [12,22] and the obligation by the European Union to follow at least an integrated form of pest management (Regulation (EU) 2016/2031).

All commercial citrus varieties are susceptible to HLB [16,34]. However, the citrus rootstock known as Flying dragon (*Poncirus trifoliata* L. Raf) and some of its hybrids have been described as HLB-tolerant species [35]. In contrast, ‘Cleopatra’ mandarin (*Citrus reshni* Hort. ex Tan) has been tested as an HLB-susceptible species [35]. Carrizo citrange (*C. sinensis* L. Osbeck x *P. trifoliata* L. Raf) is the most common rootstock in the Mediterranean basin [40], because of its cold tolerance, and despite its disadvantages: it is highly sensitive to salinity and to limestone [32,33]. Hybrids from Spanish breeding programs such as Forner-Alcaide no. 5 (*C. reshni* Hort. ex Tan x *P. trifoliata* L. Raf) and Forner-Alcaide no. 517 (*C. nobilis* L. Lour x *P. trifoliata* L. Raf) are resistant to CTV, *T. semipenetrans* and *Phytophthora* spp. [40,41,42]. Both are also tolerant to salinity and have a better response to limestone [43]. Although *Citrus macrophylla* Wester is another widely used citrus rootstock, it is susceptible to root asphyxia and *T. semipenetrans* [43].

Several factors influence the behaviour of *T. erytreae* when choosing a host plant: visual attraction, the emission of volatiles and physical or/and chemical stimuli [27,44,45]. In this regard, it is known that *T. erytreae* prefers to feed on *C. lemon* than on bitter orange (*C. aurantium*) [27], but more studies with a greater range of citrus species are needed. Citrus rootstock also influences psyllid choice preference, with *Citrus macrophylla* species and the hybrid Carrizo citrange being more susceptible to *T. erytreae attack* [46]. In addition, it was seen that *Diaphorina citri* Kuwayama, 1908 (Hemiptera: Liviidae), another HLB vector, used Carrizo citrange as a preferred host [47].

The biology and behaviour of *T. erytreae* on the main rootstocks used in Spanish citrus orchards has previously been studied [46]. However, it is unknown how the combination with the scion cultivar can affect it. Therefore, we have studied the survival, colonisation preference and oviposition of *T. erytreae* in different citrus rootstocks and cultivar combinations, with the aim of helping growers to choose the appropriate combination to avoid the damage caused by *T. erytreae* infestation and to increase knowledge for citrus breeding programs and new lines of research.

## 2. Materials and Methods

### 2.1. Plant Material

Six different citrus rootstocks, (1) Flying dragon (FD) (*Poncirus trifoliata* L. Raf), (2) ‘Cleopatra’ mandarin (CL) (*Citrus reshni* Hort. ex Tan), (3) Carrizo citrange (CC) (*C. sinensis* L. Osbeck x *P. trifoliata* L. Raf), (4) Forner-Alcaide no. 5 (FA5) (*C. reshni* Hort. ex Tan x *P. trifoliata* L. Raf), (5) Forner-Alcaide no. 517 (FA517) (*C. nobilis* L. Lour x *P. trifoliata* L. Raf) and (6) *C. macrophylla* Wester (CM), were evaluated in combination with six cultivars: (A) Star Ruby (*C. paradisi* Macf), (B) Clemenules (*C. clementina* Hort. ex Tan), (C) Navelina (*C. sinensis* L. Osbeck), (D) Valencia Late (*C. sinensis* L. Osbeck).), (E) Fino 49 (*C. lemon* Burm) and (F) Ortanique (*C. reticulata* Blanco x *C. sinensis* L. Osbeck).

Two-year-old persistent-insecticide-free plants were supplied by a commercial nursery (Viveros Sevilla S.L.). After a two-week acclimatization period at the ICIA facilities, plants around 100–120 cm in height were transplanted to pots (Ø = 20 cm) with a substrate compound of peat, compost and vermiculite (1:1:1). Plants were irrigated on demand and fertilized with a modified Hoagland’s solution [48]. With the aim of obtaining homogeneous young shoots and guaranteeing the establishment of *T. erytreae*, plants were pruned three weeks before the beginning of each experiment. Trials were spaced over 6 months.

### 2.2. Source of Insects

The *T. erytreae* individuals belonged to the colony described in Quintana et al. [46,49] of the Instituto Canario de Investigaciones Agrarias (ICIA, Santa Cruz de Tenerife, The Canary Islands, Spain), originally collected from sweet orange trees in Tegueste (Santa Cruz de Tenerife, The Canary Islands, Spain). Psyllids were raised on pesticide-free lemon (cv. Eureka grafted on CM) and sweet orange (cv. Lane Late on CC) plants inside insect-proof facilities under controlled conditions (20 ± 5 °C, RH > 70%, 16:8 h (L:D) photoperiod). Plants were pruned regularly to stimulate the emergence of new shoots.

In order to obtain 72 h old individuals, two cages were needed: one transitable for oviposition (2.9 m × 2.70 m × 2.9 m), and another one for nymphal development (1.4 m × 1.0 m× 1.0 m). Ten plants were placed in the oviposition cage, where 100 psyllids were released (sex ratio 1:1) during 2 d. Then, they were removed with a handheld aspirator and the plants were transferred into the nymphal development cage.

### 2.3. Experimental Design

The experiments were conducted between 2019 and 2021, in a greenhouse with mesh walk-in cages (2.9 m × 2.7 m × 2.9 m) at the ICIA facilities (Valle de Guerra, Santa Cruz de Tenerife, Spain). The experiments were spaced over time, but not the replicates of each one. Appendix A shows the timeline of the experiments. The experimental design was fully randomized, in which each plant was the experimental unit.

#### 2.3.1. *Trioza erytreae* Survival Assay

Groups of eight 72 h old *T. erytreae* adults (4♂ and 4♀) were chosen from the colony (described in Section 2.2) and randomly placed on tender shoots of each plant inside a 160 µm mesh bag (dimensions: 15 cm × 10 cm). After psyllid confinement, the adults of *T. erytreae* were monitored six times: 1, 3, 7, 17, 24 and 31 days after release. The citrus combinations tested in this trial are shown in Table 1. Four replicates were used per combination.

The percentage of survival and mortality was calculated using the following formulas:$$Survival\left(\%\right)=\frac{Tns}{T0}\cdot 100;Mortality\left(\%\right)=\frac{Tnm}{T0}\cdot 100$$

-Tns = Alive individuals per assessment period and citrus combination.-Tnm = Dead individuals per assessment period and citrus combination.-T0 = Individuals at the beginning of the experiment and citrus combination.

#### 2.3.2. *Trioza erytreae* Settlement Preference Assay

This experiment was performed under multi-choice conditions, where 29 citrus combinations were offered to *T. erytreae* with the aim of evaluating its settle preference. One plant of each citrus combination (Table 2) was randomly placed in a 160 µm mesh walk-in cage (2.9 m × 2.7 m × 2.9 m). Four cages were used, at the same time, as replicates. A group of 200 adults of *T. erytreae* (72 h old, sex ratio 1:1) was chosen from the colony (described in Section 2.2) and placed in each cage. The psyllids were released 1 m above the plant canopy in the centre of the cage. They were confined for 21 d, while preference on the different citrus combinations was evaluated six times (1, 3, 7, 14 and 21 d after psyllid release), counting adult individuals on the leaf surface. The counting was undertaken, avoiding disturbing them, at 8.00 am.

#### 2.3.3. *Trioza erytreae* Oviposition Assay

Two pairs of 72 h old adults of *T. erytreae* (2♂ and 2♀) were chosen from the colony (described in Section 2.2) and placed on young shoots of similar lengths, approximately 4 to 5 cm, from each citrus combination (Table 3) inside a 160 µm-mesh bag (15 cm × 10 cm). They were confined for 3 d. After this period, adults were removed and the number of eggs per shoot was counted using a stereoscopic microscope. Four replicates were used per citrus combination, and those replicates in which gravid females had died were discarded.

### 2.4. Data Analyses

The statistical software R (R version 4.0.3, R Foundation for Statistical Computing, Vienna, Austria) and the integrated development environment R-Studio (R-Studio version number 2023.03.1 Build 446, Posit Software, PBC, Boston, MA, USA) were used for the data analysis. Kaplan–Meier curves were used for the analysis of the total number of adults (alive and dead) monitored during six time-points in different citrus combinations of scion varieties and rootstocks, and their comparison was undertaken using the Log Rank equality test. The normality of *T. erytreae* oviposition data in different citrus combination was tested using the Shapiro–Wilk test. Those non-Gaussian variables were transformed by logarithm or square root. Comparison between rootstocks were made with an ANOVA-test and T-student test. *p*-values below 0.05 were considered statistically significant.

## 3. Results

### 3.1. Trioza erytreae Survival

The influence of six citrus rootstocks grafted onto ‘Ortanique’ in the *T. erytreae* survival was monitored six times, 1, 3, 7, 17, 24 and 31 d after psyllid release. Mortality and survival results (frequency (n) and percentage (%)) are shown in Appendix A. The survival rate decreased during the experiment, with lower survival percentages obtained from day 17 onwards. The citrus rootstocks in which ‘Ortanique’ recorded some *T. erytreae* individuals at the end of the trial were FD, FA5 and CC. In contrast, no alive psyllids were found in CC and FA517 from day 24. From day 1, *T. erytreae* showed the highest mortality (40.6%) on the FA517 rootstock.

To estimate the survival function in ‘Ortanique’ grafted on different rootstocks over time, Kaplan–Meier curves were performed (Figure 1). The Kaplan–Meier estimate for the ‘Ortanique’ cultivar is shown in Appendix A. Significant differences were obtained between rootstocks (χ^2^: 10.467; df: 4; *p*-value: 0.033), with FA517 allowing a lower survival of *T. erytreae* in comparison with FA5 (*p*-value: 0.036).

The survival of *T. erytreae* during the same six follow-up times was also evaluated in different combinations with the rootstocks CC and CM. Results and statistical analyses of survival and mortality (frequency (n) and percentage (%)) are shown in Appendix A. No psyllids were found on CC grafted with ‘Valencia Late’ from day 17, with ‘Clemenules’ and ‘Ortanique’ from day 24 and with ‘Fino 49’ at the end of the trial (day 31). Although some psyllids were seen in ‘Navelina’ on CC, only 3% of individuals survived to the end of the trial. In contrast, CM maintained live individuals at the end of the trial in all cultivars except ‘Clemenules’, with survival rates up to 25% (‘Fino 49’). ‘Navelina’ on CM showed a 12.5% survival of *T. erytreae* at the end of the trial. The survival percentage from day 1 to 7 was higher than 50% in all citrus combinations, except in ‘Valencia Late’ grafted onto CC, in which only 34% of individuals survived.

Figure 2 shows the Kaplan–Meier curve of *T. erytreae* survival in different scion varieties grafted onto CM and CC. No significant differences between citrus rootstocks were obtained in ‘Clemenules’, ‘Navelina’ and ‘Fino 49’ (Figure 2, graphs A, B and D, respectively). However, in the case of ‘Valencia Late’ (Figure 2C), CC allowed a lower survival of *T. erytreae* than CM (χ^2^: 7.611; df: 1; *p*-value: 0.006).

### 3.2. Trioza erytreae Preference

The settling preference of *T. erytreae* was studied in 29 citrus combinations at 1, 3, 7, 14 and 21 d after psyllid release. The number of psyllids found per plant decreased during the monitoring period. We recovered 35.4%, 33.4%, 21.6%, 14.0% and 8.6% of the total individuals released at the beginning of the experiment on days 1, 3, 7, 14 and 21, respectively. Appendix A shows the number of psyllids (mean ± SE) seen per combination and monitoring time. On day 1, psyllids per combination ranged from 0.0 ± 0.0 to 5.3 ± 2.3 (mean ± SE), corresponding to ‘Clemenules’ grafted onto FA517 and ‘Ortanique’ grafted onto CC, respectively. Generally, the settlement preference did not follow a clear trend during the five monitoring periods in ‘Ortanique’, ‘Star Ruby’ and ‘Valencia Late’. At the end of the assay, the number of psyllids counted in ‘Ortanique’ grafted onto CC and in ‘Valencia Late’ grafted onto FD were 3.3 ± 2.9 and 2.3 ± 1.3 (mean ± SE), respectively, while in most of the combinations no psyllids were seen.

### 3.3. Trioza erytreae Oviposition

The oviposition of *T. erytreae* from two psyllid pairs confined to 17 citrus combinations was evaluated after 3 d. Eggs laid by *T. erytreae* females on each combination were counted and data were statistically compared between rootstocks (Table 4).

No significant differences were obtained among rootstocks grafted with ‘Clemenules’, ‘Navelina’, ‘Valencia Late’ and ‘Ortanique’. However, differences were seen when ‘Fino 49’ was grafted onto different rootstocks (F: 6.474; df: 3; *p*-value: 0.004), with *T. erytreae* able to lay more eggs in CC than in FD or FA5.

## 4. Discussion

*Tryoza erytreae* has rapidly spread along the Atlantic coast of the Iberian Peninsula. The aggressiveness of its symptoms and the possibility of the arrival of HLB threaten European citrus production. Although many resources have been used to directly reduce the psyllid population, including the use of biocontrol, pesticides, traps, etc., few studies consider the role of citrus species and even the scion–rootstock combinations. It is not known which scion–rootstock combination influences the biology or behaviour of *T. erytreae* for better or worse regarding survival, preference and oviposition. In another words, is it possible that *T.* erytreae behaves differently with the same scion variety grafted onto different rootstocks?

To answer this question, we first analysed the survival of *T. erytreae* on five rootstocks grafted with ‘Ortanique’, revealing that all of them were suitable for psyllid subsistence. However, FA517, followed by CC, obtained the lowest survival, and FD, FA5 and CM had the highest values. It is noteworthy that on day 3 after psyllid release, more than half of the individuals survived in all citrus combinations, but thereafter, no more than 15.6% of individuals survived. This decline in survival could have been accelerated due to the life cycle of *T. erytreae*, which is highly influenced by temperatures and minimum vapor pressure [50,51]. However, we also suggest that the scion variety influences psyllid biology and behaviour. ‘Ortanique’ is a natural hybrid of *C. sinensis* and *C. reticulata* discovered in Jamaica, which is used for its extremely sweet flavour balanced with acidity [52,53]. It is known that *T. erytreae* prefers to feed on *C. lemon* than on bitter orange (*C. aurantium*) [27], and in the field shows a preference for lime, lemon and sweet orange trees [54,55,56], but there are no works on ‘Ortanique’. Interestingly, in our work, ‘Ortanique’ on FD allowed the highest survival of more individuals at the end of the experiment, while on FA517 it obtained the lowest survival values. These data contrast with those obtained by Hernández-Suárez et al. [46], where CC, CM and CL were the citrus rootstocks with the highest survival percentages, compared to FA5, FA517 and FD. Since this study was conducted under similar conditions, we can conclude that the grafted scion variety influences the biology of *T. erytreae*. On the other hand, FD is an interesting rootstock because it has been shown to be tolerant to HLB [35].

Secondly, a comparison was made between two citrus rootstocks, CC and CM, when grafted with five different scion varieties. During the first count, there were no differences among them, except in the case of ‘Valencia late’, in which CC presented a lower survival percentage than CM. Considering all the results obtained at the end of the experiment, it seems that CM allowed the survival of more individuals than CC, coinciding with Hernández-Suárez et al. [46]. As mentioned in the introduction, CC is the most used rootstock in the Mediterranean citrus industry [46]. It is a trifoliate orange hybrid which has been cited as being moderate tolerant to HLB [34]. Although CM is also a widely used rootstock, we believe that its use should be limited, with the aim of reducing the risk of arrival and spread of HLB to Mediterranean citrus orchards, due to it having been reported as highly susceptible to HLB [57].

Survival experiments were carried out under no-choice conditions, giving information on subsistence in different species without the possibility of obtaining other food sources. However, the most common situation in field conditions is that psyllids choose their feeding source freely [49,54,55,56]. Therefore, a multichoice experiment was conducted with 29 citrus combinations, in order to study the preferences of settlement. A clear preference was not observed for landing on any plant, suggesting that the conditions of the experiment could have influenced the results, since so many citrus combinations were offered to *T. erytreae* in a big transitable cage where 200 psyllids were released. These conditions were previously and satisfactorily used by Hernández-Suárez et al. [46], demonstrating a clear preference for CM during the first year of the experiment but offering only six plants in the free-choice trial. CM was previously reported as a highly preferred species by *D. citri*, being used for psyllid rearing [58]. However, Urbaneja-bernat et al. [47] found that *D. citri* preferred CC within the rootstock offered. Our results could be considered as a preliminary test, and we believe that more experiments should be carried out to improve the experimental setup; it is strongly recommended to reduce the amount of citrus combinations offered. A marked effect by some combinations was observed in the oviposition experiment, where *Trioza erytreae* significantly preferred to oviposit on ‘Fino 49’ when it was grafted onto CC.

Recently, in 2021, *D. citri* was detected on *C. reticulata* and *C. sinensis* trees, in groves within a limited area, in central Israel, and in 2023 in Cyprus [23] Although HLB was not detected and eradication efforts are underway, the threat of its spread seems increasingly possible. Finding a rootstock capable of dealing with HLB and its vectors is essential in management plans. The present work helps to understand the role of citrus combination in the biology and behaviour of *T. erytreae*, essential to further unravelling the epidemiology of the tritrophic complex in which this psyllid vector is involved. Additionally, this knowledge may help to reduce the psyllid population by using citrus combinations that are less suitable to *T. erytreae* development.

## Figures and Tables

**Figure 1 insects-15-00363-f001:**
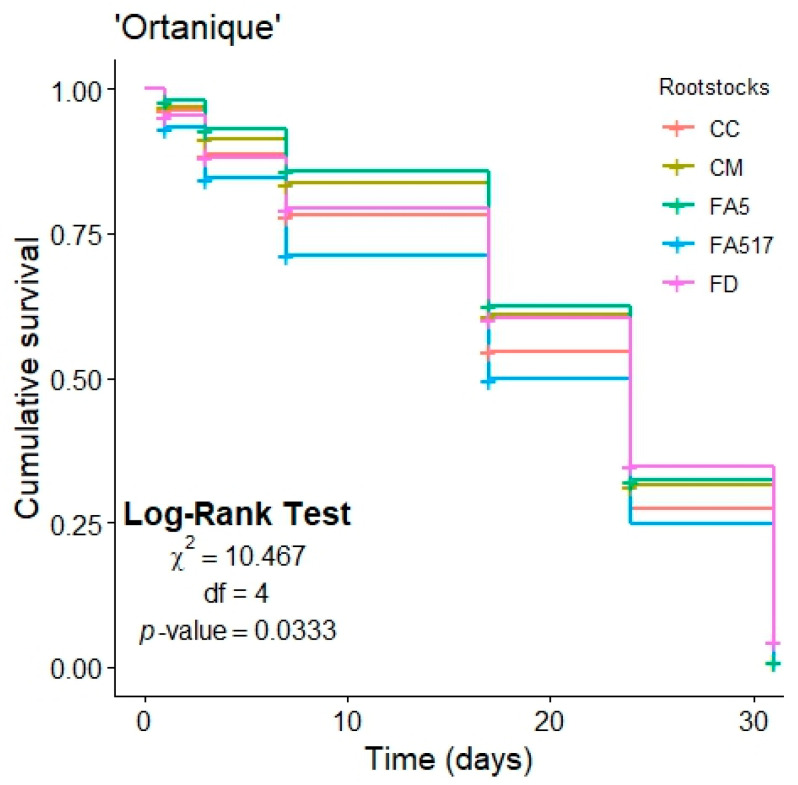
Kaplan–Meier curves of *T. erytreae* survival in ‘Ortanique’ grafted onto five citrus rootstocks (FD, CC, FA5, FA517 and CM) at six follow-up times (1, 3, 7, 17, 24 and 31 d after psyllid release). CC: Carrizo citrange; CM: *Citrus macrophylla*; FA5: Forner-Alcaide no. 5; FA517: Forner-Alcaide no. 517; FD: Flying dragon. Cumulative survival is expressed as a decimal.

**Figure 2 insects-15-00363-f002:**
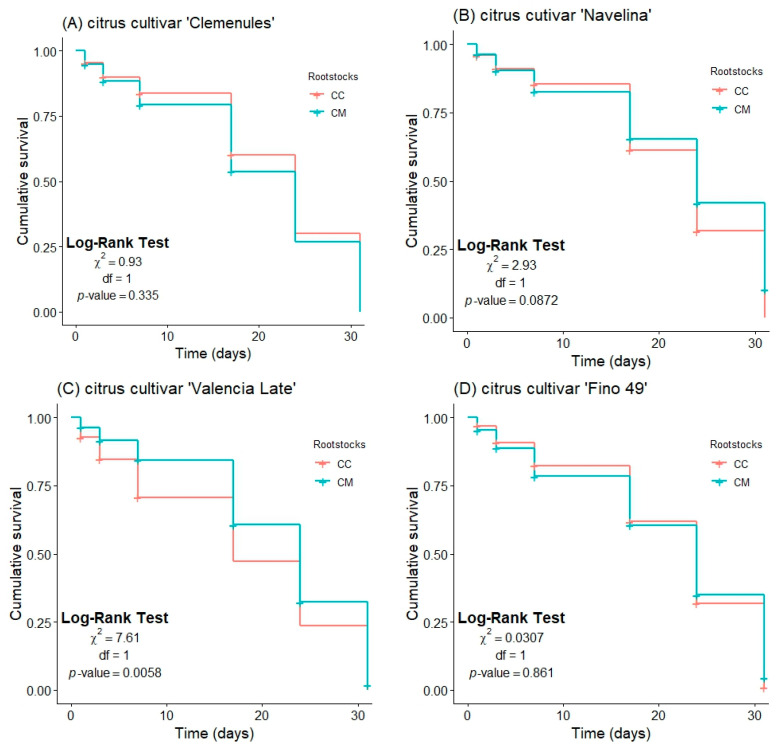
Kaplan–Meier curves of *T. erytreae* survival in the following citrus cultivars: ‘Clemenules’ (**A**), ‘Navelina’ (**B**), ‘Valencia Late’ (**C**) and ‘Fino 49’ (**D**) grafted onto two citrus rootstocks (CC, Carrizo citrange and CM, *Citrus macrophylla*) during six follow-up times (1, 3, 7, 17, 24 and 31 d after psyllid release). Cumulative survival is expressed as a decimal.

**Table 1 insects-15-00363-t001:** Citrus combinations used in *T. erytreae* survival assay.

Rootstock	Cultivar				
	Clemenules	Navelina	Valencia Late	Fino 49	Ortanique
FD	NT	NT	NT	NT	x
CC	x	x	x	x	x
FA5	NT	NT	NT	NT	x
FA517	NT	NT	NT	NT	x
CM	x	x	x	x	x

NT: Citrus combination not tested. x: Tested combination. FD: Flying dragon; CC: Carrizo citrange; FA5: Forner-Alcaide no. 5; FA517: Forner-Alcaide no. 517; CM: *Citrus macrophylla*.

**Table 2 insects-15-00363-t002:** Citrus combinations used in *T. erytreae* preference assay.

Rootstock	Cultivar					
	Star Ruby	Clemenules	Navelina	Valencia Late	Fino 49	Ortanique
FD	x	x	NT	x	x	x
CL	x	x	x	x	NT	NT
CC	x	x	x	x	x	x
FA5	x	x	x	x	x	x
FA517	NT	x	x	NT	NT	x
CM	NT	x	x	x	x	x

NT: Citrus combination not tested. x: Tested combination. FD: Flying dragon; CL: Cleopatra mandarin; CC: Carrizo citrange; FA5: Forner-Alcaide no. 5; FA517: Forner-Alcaide no. 517; CM: *Citrus macrophylla*.

**Table 3 insects-15-00363-t003:** Citrus combinations used in *T. erytreae* oviposition assay.

Rootstock	Cultivar				
	Clemenules	Navelina	Valencia Late	Fino 49	Ortanique
FD	NT	NT	x	x	x
CC	x	x	x	x	x
FA5	NT	NT	x	x	x
FA517	NT	NT	NT	NT	x
CM	x	x	x	x	x

NT: Citrus combination not tested. x: Tested combination. FD: Flying dragon; CC: Carrizo citrange; FA5: Forner-Alcaide no. 5; FA517: Forner-Alcaide no. 517; CM: *Citrus macrophylla*.

**Table 4 insects-15-00363-t004:** Number of eggs (mean ± SE) laid in each citrus combination.

Rootstock	Scion Cultivar				
	Clemenules	Navelina	Valencia Late	Fino 49	Ortanique
FD	NT	NT	60.8 ± 36.8 a	22 ± 9.5 bc	74.4 ± 29.3 a
CC	79.6 ± 23.2 ns	112.2 ± 40.9 ns	58.6 ± 23.6 a	145 ± 40 a	71.2 ± 17.8 a
FA5	NT	NT	77.6 ± 19.6 a	32.4 ± 7.3 bc	100.8 ± 23.9 a
FA517	NT	NT	NT	NT	154.2 ± 25.6 a
CM	87.2 ± 35.7 ns	64 ± 26.2 ns	61.6 ± 21.6 a	43.6 ± 13.3 ab	90.6 ± 27.2 a

NT: Citrus combination not tested. Different letters mean significant differences among rootstocks (*p*-value < 0.05). ns: no significant differences were found.

## Data Availability

The raw data supporting the conclusions of this article will be made available by the authors on request.

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
