# Peer review of "The Combination of Citrus Rootstock and Scion Cultivar Influences Trioza erytreae (Hemiptera: Triozidae) Survival, Preference Choice and Oviposition"

_insects, 2024, doi:10.3390/insects15050363_

Round 1

Reviewer 1 Report

Comments and Suggestions for Authors

Line 151 (following equations): is more appropriate (following formulas) or (following expressions).

Line 153: What do (1) and (2) mean in formulas?

 2.3.2. Trioza erytreae preference assay

Line 162: The preference cannot be monitored but evaluated by the number of individuals on the leaves surface of the several combinations.

Line 163: in figure 1 (at six follow-up times (1, 3, 7, 17, 24 and 31 d after psyllid  release).

In this section it is important to give some information about the randomness or not of the assay, how the plants have been placed and the distance between plants inside the cage. How are insects released into the cage in relation to plants? Since the preference is studied in the adult stage, how do you ensure that one insect does not fly during the counts to other plants and is not counted twice? What is sex ratio? (100 males and 100 females)?

All this information is important in preference assays.

 Results

 3.2. Trioza erytreae preference

 Line 188: In all observations, a decrease in the number of individuals was observed; What's different about the 17th day?

Line 191: FA517 had a high mortality rate from the beginning (40.6 % in day 1). I really don’t understand this sentence.

Line 206: (No psyllids were found on CC grafted with `Valencia Late’ from day 17, with `Clemenules’ and `Ortanique’ from day 24 and …). Question: If there are no psyllids, why in Kaplan-Meier curves there is cumulative survival different from 0. In these cases cumulative survival it shouldn't be 0?

 3.2. Trioza erytreae preference

 Line 224: (Settling preference) This expression was not defined in the MM section. The meaning should be indicated.

It should be clearly explained in the MM section how the assay was conducted. Are all the combinations placed inside the cages or several trials have been carried out, each with its own cultivar? The figures should correspond to the way the test assay was conducted.

There is no statistical analysis to compare the various combinations.

Figure 3 has a very low quality. It is difficult to analyse the figure.

This whole section is very confusing, and it is not clear how it relates to the respective section of MM. The text related to figure 3 is poorly structured and the lack of statistical treatment does not allow an adequate understanding of this section. This, together with the revisions made to the methodology, makes this section inconsistent. I think it needs a major methodological and descriptive revision. Behavioural studies are already complicated in the laboratory with well-controlled conditions. When the discriminating stimuli are not sufficiently distinct, they make these assays unproductive.

 Discussion

Line 300: You must specify what do you mean by “different behaviour”.

Line 301-306 – How can we verify this? There is no statistical analysis. On what days does this happen? If the goal was to show the differences between cultivars for the CC rootstock, why not to make a figure only with this rootstock.

Author Response

Manuscript ID insects-2960820

The combination of citrus rootstock and scion cultivar influences Trioza erytreae (Hemiptera: Triozidae) behaviour: survival, preference choice and oviposition.

“Point by point” answer to reviewer’s comments (attached to the resubmitted manuscript)

Reviewer 2 Report

Comments and Suggestions for Authors

Excellent study examining important aspects of psyllid/host interactions.  Well written and conducted experiment. I only have a couple of minor comments

Can you describe in a little bit more detail about the observations (line 163-164). You did visual counts of all psyllids sitting on the leaves? Where the counts done at the same time of day for each of your observation dates? As you stated, there are four replicates so you did the counts for each of the replicates within the same time period?

The settling or preference assay is a challenge because there could have been other factors (side of cage, lighting, aim movement) that might have influenced psyllid movement in the cage.  It is a difficult thing to assess and your methods are as good as any other approach.

Figure 3 seems a bit blurry

Author Response

Manuscript ID insects-2960820

The combination of citrus rootstock and scion cultivar influences Trioza erytreae (Hemiptera: Triozidae) behaviour: survival, preference choice and oviposition.

“Point by point” answer to reviewer’s comments (attached to the resubmitted manuscript)

Reviewer 2 (R2)

R2

Line

Comment

Reply from the authors

A

163-164

Can you describe in a little bit more detail about the observations (line 163-164). You did visual counts of all psyllids sitting on the leaves? Where the counts done at the same time of day for each of your observation dates? As you stated, there are four replicates so you did the counts for each of the replicates within the same time period?

The settling or preference assay is a challenge because there could have been other factors (side of cage, lighting, aim movement) that might have influenced psyllid movement in the cage.  It is a difficult thing to assess and your methods are as good as any other approach.

The Materials and Methods of this experiment was improved in the manuscript, and a new figure was added to Supplementary Material (Figure S2).

Psyllid were counted while they were settling on the leaves, avoiding disturbing them. Monitoring was done at 8.00 am. At this time of the day psyllids are not as active as, for example, at noon.

The replicates were not spaced over time, four transitable cage were used at the same time (see Figure S2).

The plants were randomly located into the cages with the aim to prevent the influence of light incidence, orientation or box edge effect.

B

Figure 3 seems a bit blurry

We agree with reviewer opinion, and so we have modified the results text writing in order to clarify this section.

We removed the Figure 3, and we have added a new table in Supplementary Material (Table S3), which hopefully could help for the understanding. 

Reviewer 3 Report

Comments and Suggestions for Authors

This is a review of “The combination of citrus rootstock and scion cultivar influences Trioza erytreae (Hemiptera, Triozidae) behaviour: survival, preference choice and oviposition.”

Line 20) control of the disease

Line 21) no. plant material tolerant to the psyllid will probably only increase disease spread. The tolerant plant will harbor the disease and allow more psyllids that will become infected. Did you intend to say resistant to the psyllid, or at least having some resistance?

Line 27) indistinctly is the wrong word. I am not sure what the right word is. Consider deleting sentence.

Line 30) Transmission is direct.

Line 37) delete “of preference”

Line 39) grafted onto, not grafted with. (and again in line 40)

Line 52) delete “and prolific”

Line 53) Make sure there is a space between the period at the end of a sentence and the start of a new sentence. The space is missing before Although.

Line 57) delete “those”

Line 62) It is also present in the gut, gut lining, and haemolymph.

Line 95) greater range of

Line 95) delete “to the choice of psyllid.”

Line 118) How did you determine pesticide-free? Also, what do you mean? Generally, nurseries cannot produce plants that have never been sprayed with any pesticide. Do you mean any pesticide, no oil, or soap?

Line 123) So about how much time from when you got the plants from the nursery and when they were used in experiments?

Line 141) How were replicates spaced over this three-year interval? Was time a blocking variable? Did you do the survival assay in 2019, and the preference assay in 2020, and the oviposition assay in 2021?

Line 153) This looks like a single equation and as such it makes no sense. Put some separation between the two equations.

Line 153) Is there a real point? Survival % = 100 – mortality % or that is how it should work.

Table 2) in foot note: combination not tested (same problem in other tables)

Line 182) After transformation, did you test to see if the models fit? Was there enough data to reliable test for normality? Is normality of the observed data an assumption of the model? In many models the assumption of normality is not on the raw data.

Line 188) “The survival evolution” needs to be remove or rephrased. There is no survival evolution, at least not in the Darwin sense of evolution. In figure 1 I do not see anything special happening at day 17 onwards.

Line 190-192) I do not see Table S1 in Figure 1. For FA517 in Table S1 I agree that mortality in Day 1 was 40.6%. However, in Figure 1 survival was above 80% in day 1. Please explain.

Line 214) I just do not see that in the figures. Why the mismatch between table and figures? How about adding another few calculations to Tables S1 and S2 so that it is clear how tables and figures match up, or adding some brief text in the supplement to explain? I tried looking at a paper (https://www.ncbi.nlm.nih.gov/pmc/articles/PMC3059453/) and had no trouble there.

Line 225) decreasing evolution is not a concept. Are people turning back into fishes? Probably just delete sentence.

Line 226) Rather than “It was possible …” try “ We recovered 35.4%, 33.4%, 21.6%, 14.0%, and 8.6% of the total individuals released at the beginning of the experiment on days 1, 3, 7, 14, and 21 respectively.”

Figures) Always include units for x and y axes. It can be “Cumulative survival (%)” or “Cumulative survival (percent)” or “Percent cumulative survival” or in your case it looks more like “Cumulative survival probability” and of course “Time in days” or “Time (D)”, or something like that.

Figure 3) The figure is hard to follow. Most of the error bars overlap.

Line 239) No. CC is the purple dots. On Orthanique on day 1 there were three psyllids not “more than 5.”

Line 242) In general use past tense for all verbs in the results. All of the results were in the past at the time of writing.

Figure 3) There were no zero values in any treatment? Every plant had at least one psyllid??

Line 258) irevise this sentence, possibly breaking it into two smaller sentences.

Line 259) biocontrol, not bio-controlers. You could say biological control.

Line 259) Pesticide would be more concise relative to “chemical active ingredients” though you could argue that people mostly think of synthetic pesticides when one states “pesticides” as opposed to the full range of bioactive molecules.

Line 263) It is certain that this takes place, and I would give that answer even without the presented data. It is easy to say there will be a difference, much harder to identify which combinations are better or worse than others.

Line 265) Maybe. You could have had a couple of other control type treatments. No food or water, for a starved control. My guess is that they will live 2 to 4 days in this treatment. Water alone could be a control or use a non-host. Recent research has shown that many hemipterans can use non-hosts to survive adverse conditions. Gut content analysis and looking at survival curves are two approaches. A third is to use electropenetrography. A good non-host might extend life out to 8 to 10 days. Those sorts of controls would provide context regarding psyllid subsistence. At the other extreme, what would the data look like on a great host? Could any of the hosts used herein serve as a long term rearing host? These additional experiments are not a requirement, but they would have provided useful context.

Line 268) use period rather than comma.

Line 268) Rephrase to be clearer. It looks like survival was above 50% in some cases at 17 days. If on day 3  more than 50% survived but thereafter (=days 7 onwards) survival was less than 15.6%, how do you get above 50% survival at 17 days?

Line 285) were no differences

Line 298) psyllids choose their feeding source

Line 300) make clearer

Line 321) Please read the manuscript carefully. Exactly how would using citrus combinations that are suitable for T. erytreae development going to help reduce psyllid populations?

Comments on the Quality of English Language

The English had some issues. In a few places they said things that I am fairly sure they did not mean, or they need to add some additional explanation to make the text clearer. There were some poor word choices.

Author Response

(The authors gave the same response as above.)

Round 2

Reviewer 1 Report

Comments and Suggestions for Authors

Line 169 “1, 3, 7, 17 and 21”: In the results is “1, 3, 7, 14 and 21”,

Line 267 – “survival” is not a behavioural parameter but a biological one. Also on line 288. In line 277 it is not clear if behaviour is related to survival.

Line – 242 However, it seems that ‘Clemenules’ and ‘Navelina’ grafted on CC, and ‘Fino 49’ on FD were combinations preferred by T. erytreae, at least during the first 7 days after psyllid release.

Line 306 - However, some combinations seem to be more suitable for the psyllid settlement: ‘Clemenules’ and ‘Navelina’ grafted on CC, and ‘Fino 49’ on FD. This suggest that there is an influence of scion variety on the settlement of T. erytreae because when ‘Fino 49’ is grafted on CC, the number of psyllids (mean±SE) found on the shoots range 309from 0.0±0.0 to 0.5±0.3 during the monitoring period.

It is very dubious to attempt to draw conclusions from this essay for the following reasons:

The experimental setup is not the most suitable:

1.                       It is done under natural light (apparently), so the light conditions are not homogeneous inside the cage and this can have a obvious effect on the insects' choice of plants. Even if in the different cages the arrangement is different, this may not be enough to exclude this effect.

It could have been more functional to use fewer different combinations per cage, with 2-3 plants per combination in the same cage, placing the plants in different positions.

2.                       The amount of insects on each plant is so low that, if some insects fly to other plants during the count (it's not unlikely) this could have a strong effect on the mean.

3.                       There is no statistical treatment and therefore it is not possible to say whether or not there is a significant difference between two means.

I advise you to consider this test as a preliminary study to see if there is any marked effect by some combinations, concluding only that ,this effect was not observed.

Author Response

Manuscript ID insects-2960820

The combination of citrus rootstock and scion cultivar influences Trioza erytreae (Hemiptera: Triozidae) behaviour: survival, preference choice and oviposition.

Round 2

“Point by point” answer to reviewer’s comments (attached to the resubmitted manuscript)

Reviewer 1 (R1)

R1

Line

Comment

Reply from the authors

A

169

“1, 3, 7, 17 and 21”: In the results is “1, 3, 7, 14 and 21”,

The error in the Materials and Methods section was corrected: Line 170: 1, 3, 7, 17, 24 and 31 days.

The manuscript was carefully reviewed and we found no further errors. It is important to note that the survival assay lasted 31 days, being monitored at 1, 3, 7, 17, 24 and 31 days. In the case of the preference assay, it lasted 21 days, being monitored at 1, 3, 7, 14 and 21 days.

B

267, 288

“survival” is not a behavioural parameter but a biological one. Also on line 288.

We agree with the reviewer suggestion, several corrections were made in the text.

C

277

In line 277 it is not clear if behaviour is related to survival.

Same reply to comment “C”.

D

242

However, it seems that ‘Clemenules’ and ‘Navelina’ grafted on CC, and ‘Fino 49’ on FD were combinations preferred by T. erytreae, at least during the first 7 days after psyllid release.

Same reply to comment “E”.

This sentences was deteled from the manuscript.

R1

Line

Comment

Reply from the authors

E

306

However, some combinations seem to be more suitable for the psyllid settlement: ‘Clemenules’ and ‘Navelina’ grafted on CC, and ‘Fino 49’ on FD. This suggest that there is an influence of scion variety on the settlement of T. erytreae because when ‘Fino 49’ is grafted on CC, the number of psyllids (mean±SE) found on the shoots range 309from 0.0±0.0 to 0.5±0.3 during the monitoring period.

It is very dubious to attempt to draw conclusions from this essay for the following reasons:

The experimental setup is not the most suitable:

1.       It is done under natural light (apparently), so the light conditions are not homogeneous inside the cage and this can have a obvious effect on the insects' choice of plants. Even if in the different cages the arrangement is different, this may not be enough to exclude this effect.

2.       It could have been more functional to use fewer different combinations per cage, with 2-3 plants per combination in the same cage, placing the plants in different positions.

3.       The amount of insects on each plant is so low that, if some insects fly to other plants during the count (it's not unlikely) this could have a strong effect on the mean.

4.       There is no statistical treatment and therefore it is not possible to say whether or not there is a significant difference between two means.

I advise you to consider this test as a preliminary study to see if there is any marked effect by some combinations, concluding only that , this effect was not observed.

We appreciate the reviewer's point of view, so we have followed her/his advice. We have changed the Results and Discussion sections to avoid drawing the wrong conclusions from our results.

Discussion section:

Therefore, a multichoice experiment was conducted with 29 citrus combinations, in order to study its preference of settlement. It was not observed a clear preference for landing on any plant, suggesting that the condition of the experiment could have influenced the results, since so many citrus combinations were offered to T. erytreae in a big transitable cage where 200 psyllids were released. These conditions were previously and satisfactorily used by Hernández-Suárez et al. [46] obtaining a clear preference for CM during the first year of the experiment but offering only six plants in the free-choice trial. CM was previously reported as a high preferred species by D. citri, being used for psyllid rearings [58]. However, Urbaneja-bernat et al. [47] obtained that D. citri prefers CC within the rootstock offered.  Our results could be consider as a preliminary test, and we believe that more experiments should be done improving the experimental setup, where it would be very recommended to reduce the amount of citrus combinations offered.